# Interventions to Vaccinate Zero-Dose Children: A Narrative Review and Synthesis

**DOI:** 10.3390/v15102092

**Published:** 2023-10-14

**Authors:** Erin A. Ingle, Priyanka Shrestha, Aparna Seth, Mathias S. Lalika, Jacinta I. Azie, Rena C. Patel

**Affiliations:** 1Strategic Analysis, Research & Training (START) Center, University of Washington, Seattle, WA 98195, USA; pshres08@uw.edu (P.S.); aseth2@uw.edu (A.S.); mathiaslalika@gmail.com (M.S.L.); jazie@uw.edu (J.I.A.); renapatel@uabmc.edu (R.C.P.); 2Department of Medicine, University of Alabama at Birmingham, Birmingham, AL 35294, USA

**Keywords:** vaccination, zero-dose children, interventions, barriers

## Abstract

Zero-dose children, or children who have not received any routine vaccination, are a priority population for global health policy makers as these children are at high risk of mortality from vaccine-preventable illnesses. We conducted a narrative review to identify potential interventions, both within and outside of the health sector, to reach zero-dose children. We reviewed the peer-reviewed and grey literature and identified 27 relevant resources. Additionally, we interviewed six key informants to enhance the synthesis of our findings. Data were organized into three priority settings: (1) urban slums, (2) remote or rural communities, and (3) conflict settings. We found that zero-dose children in the three priority settings face differing barriers to vaccination and, therefore, require context-specific interventions, such as leveraging slum health committees for urban slums or integrating with existing humanitarian response services for conflict settings. Three predominant themes emerged for grouping the various interventions: (1) community engagement, (2) health systems’ strengthening and integration, and (3) technological innovations. The barriers to reaching zero-dose children are multifaceted and nuanced to each setting, therefore, no one intervention is enough. Technological interventions especially must be coupled with community engagement and health systems’ strengthening efforts. Evaluations of the suggested interventions are needed to guide scale-up, as the evidence base around these interventions is relatively small.

## 1. Introduction

“Zero-dose children”, a term used for those children who have failed to receive even one of the routine vaccinations, specifically the first dose of diphtheria–tetanus–pertussis or DTP-1, are a priority population for global health policy makers [1,2,3]. Globally, the number of zero-dose children fell from 56.8 million in 1980 to 14.5 million in 2019 [4]. Although there has been marked progress in vaccinating zero-dose children, particularly in Asia, the absolute number of zero-dose children remains stubbornly high. Nearly half of deaths in zero-dose children are attributed to the unvaccinated status of these children. As such, 13% of children globally are at high risk of dying from vaccine-preventable diseases due to lack of immunity [5,6,7]. 

In 2020, 16.6 of the 17 million zero-dose children lived in low- and middle-income countries (LMICs)—with 71% of them living in middle-income countries and 26% in low-income countries [8]. Specifically, more than 65% of zero-dose children lived in 10 countries: India, Nigeria, Democratic Republic of the Congo, Pakistan, Ethiopia, Indonesia, Philippines, Angola, Mexico, and Brazil [8]. Within these countries, marked variations in the geographic distribution of zero-dose children exist, with a high prevalence observed in remote rural, low-resource urban, and conflict settings [9,10,11,12]. 

The COVID-19 pandemic has further aggravated the concern for zero-dose children, especially in LMICs [13,14]. According to the Global Alliance for Vaccine and Immunization (Gavi), in Gavi-eligible countries alone, the number of zero-dose children has soared by approximately 30% during the pandemic. This increase led to a 4% decline in global vaccination coverage in 2020 [14]. Ongoing ramifications from the pandemic are likely to lead to additional losses in the prior gains in reaching zero-dose children.

Reaching zero-dose children is critical to both achieving equity and reducing the burden of communicable diseases globally, thus impacting a great number of lives. In recent years, key global partners such as Gavi, the World Health Organization (WHO), and the United Nations Children’s Fund (UNICEF) have undertaken substantial work to understand who zero-dose children are and where they reside [2,5,6]. The barriers to reaching these children include economic, political, and socio-cultural factors. The challenge ahead lies in identifying zero-dose children and vaccinating them, which requires gaining a deeper understanding of how to reach these communities first. Thus, we conducted a narrative review to describe the potential, often innovative, interventions from health and non-health sectors to reach zero-dose children for vaccination.

## 2. Materials and Methods

We conducted a literature review to identify innovative solutions for reaching and vaccinating zero-dose children. We searched the terms “zero-dose” and “unvaccinated” to find relevant articles and organizations working in this field (see Appendix A for full search string). Because of the limited data and sources that exist on this topic, very few relevant sources were excluded. Exclusion criteria were repetition of information, non-English language, non-LMIC settings, and publications before 2010. We also excluded sources pertaining to vaccination that were not zero-dose and DTP-specific. We included relevant articles that met the criteria and proceeded with a snowball approach by following the sources referenced in the primary materials we identified as relevant articles. We included both published and gray research. All types of evidence, including peer-reviewed publications, policy papers, and reports were included in our review if deemed relevant to the research topic. 

The last phase of our synthesis included conducting key informant interviews with the lead authors of highly relevant articles and the implementing organizations working on last-mile delivery efforts for vaccination. We used a convenience sampling method to identify potential informants. We contacted the potential informants via email, then conducted the interview via Zoom using a semi-structured interview guide (see Appendix A). We developed this interview guide based on team knowledge of research questions of interest and iteratively modified it as our interviews progressed (see Appendix A). Briefly, the interview guide included areas on (1) understanding barriers and (2) interventions to reach zero-dose children. The interview was facilitated by a primary interviewer and additional team members took notes only. We used a largely deductive coding approach to the interview notes based on a template (see Appendix A) and conducted a rapid qualitative analysis to identify relevant themes. Information from these interviews provided deeper insights into the content that was either missing or had limited evidence to fortify the arguments around the need and available innovations for zero-dose vaccinations. We used saturation of themes for guiding the number of interviews we conducted. The interviews were not recorded and did not qualify for human subjects research per the Human Subjects Division at the University of Washington.

For compiling evidence, we used Excel software to categorize the information generated from the evidence source into the following large categories based on setting type: remote rural, urban slum, and conflict settings. This categorization was based on the Equity Reference Group for Immunization (ERG) which states that in addition to urban slums and remote rural areas, approximately 40% of zero-dose children live in fragile or conflict settings, and therefore reaching these areas is critical to vaccinating every child [15]. We further categorized the type of intervention identified for each of the settings into predominant, emerging themes.

## 3. Results

In total, we identified 27 relevant materials and conducted six key informant interviews. Of the six key informants three were affiliated with non-governmental agencies including PATH and Village Reach and three were affiliated with a university. Most of the references that guided our analysis were gray research documents including reports and news articles published by non-profit organizations working in this area, particularly, UNICEF, WHO, JSI, and the Equity Reference Group for Immunization (ERG). While peer-reviewed articles describing zero-dose vaccination interventions were limited, this report refers to some relevant published research articles for each setting.

We discuss specific interventions for reaching zero-dose children in three different settings where zero-dose children are commonly found: urban slums, remote rural populations, and conflict settings. Though the prevalence of zero-dose children is not limited to only these settings, reporting on interventions specific to each setting can help stakeholders adapt these solutions to maximize reach and impact. Of note, for each setting, prior to discussing potential interventions, we briefly summarize the specific barriers that have been identified for reaching zero-dose children to provide relevant context for reporting the interventions that follow. We further classified the types of interventions for each setting into the three predominant thematic areas: (1) community engagement, (2) health systems’ strengthening and integration, and (3) technological innovation. Table 1 provides a list of more detailed examples of these interventions based on the thematic areas in each setting. While many of the interventions listed below are applicable across multiple different settings, key informants stressed that adaptation of interventions to the local context is necessary in order to be most effective in reaching zero-dose children. Furthermore, while some of these interventions have been used specifically for vaccine delivery, others have been used for the delivery of other products or were merely identified as having the potential to be used in vaccination and are, therefore, not all evidence-based. Discretion and adaptation should be used especially in regard to non-traditional evidence-based solutions.

### 3.1. Urban Slums 

#### 3.1.1. Context for Barriers

Often, in urban slums, instead of physical barriers, social factors, including a lack of caretaker’s knowledge, the mother’s lack of autonomy, and distrust of the government or public health authorities, may serve as leading barriers to vaccination for children [16,17]. Therefore, cross-cutting approaches geared towards improving vaccine service delivery and building both demand and trust among communities are vital to ensure an increase in vaccine coverage among zero-dose children in urban areas. 

#### 3.1.2. Interventions for Community Engagement 

Key informants stressed that community involvement and participation are essential to reinforcing efforts to reach zero-dose children. Particularly, gaining trust and reliability is key to mobilizing urban communities. For instance, a community’s cultural norms must be understood to tailor appropriate messages for vaccination. Examples from our review corroborate these points. In countries such as Afghanistan and Nigeria, religious and traditional leaders were involved in promoting vaccination in fragile areas and conducting door-to-door outreach vaccination programs [18,19]. These leaders received non-monetary incentives to recognize their work [19]. Moreover, co-creating interventions with community assets such as informants, influential members, and survivors of vaccine-preventable disease, such as polio survivors, enhances ownership and strengthens trust in tailoring the crucial components of the vaccination services. 

Engaging urban slum communities in the decision-making processes, specifically in identifying and prioritizing their challenges, also impacts vaccine uptake by expanding reach and community acceptance [20,21]. Children in urban slums in India have been organizing campaigns to map their neighborhoods and identify, among other things, where play spaces should be located [22]. Similar community mapping initiatives may help in identifying an ideal location or conditions of the vaccination center or time to offer the services. This intervention may also be leveraged to improve the strengthening of health systems. 

Health promotion messages using artistic measures have been shown to be effective and widely supported in urban settings, particularly in East Africa [23,24,25,26]. UNICEF, in partnership with GOAL Zimbabwe, implemented a wide-reaching outdoor art campaign to promote healthy practices to limit the spread of COVID-19 [24]. Since this approach has the potential to reach people regardless of access to technology, literacy, or language barriers, it can be leveraged to improve vaccine uptake for zero-dose children in urban areas.

#### 3.1.3. Interventions for Health Systems Strengthening and Integration 

We repeatedly found that integration of vaccination services into existing health services can help counteract certain barriers. For instance, establishing a referral system between providers of non-vaccination services and existing vaccination centers can help reach children in urban slums who already access other health services. Pakistan is employing this approach by building a referral system to ensure that unvaccinated children attending pediatric departments for non-vaccine related care are referred to vaccination centers [17,27]. Another way to increase coverage is by tailoring the location and timing of vaccination services to reach zero-dose children whose caretakers may have difficulties taking their children to vaccination centers during traditional working hours [28]. Many LMICs have adapted this approach through community mapping activities to increase vaccination coverage; for example, vaccination outreach activities in Uganda were moved to weekends, the opening hours of vaccination centers in Kenya were extended [19], and offering evening vaccination services in slum areas in Bangladesh was associated with a significant increase in vaccine uptake [29].

#### 3.1.4. Interventions for Technological Innovations in Vaccine Delivery

Poor data collection and quality impact the assessment of vaccination coverage in LMICs, making it challenging to identify zero-dose children living in urban areas [30]. Thus, innovative health monitoring initiatives are key to addressing data quality issues and a few examples stand out for urban slum communities. One effective health monitoring system is an urban immunization dashboard in India that is reaching unvaccinated children by leveraging the existing health management information systems [19]. Another similar initiative in Uganda exists where health information systems are used to identify adverse events post-vaccination and respond promptly, hence reducing the likelihood of misinformation and improving parents’ trust in vaccines [31]. In Afghanistan and Pakistan, permanent transit teams and cross-border vaccination centers were established and supplemented by geographic information systems (GIS) monitoring to improve data management and quality [17].

### 3.2. Remote or Rural Populations

#### 3.2.1. Context for Barriers 

Accessibility, poverty, weak infrastructure for data and human resources, and a lack of structural power are some factors impacting equitable vaccine coverage in remote or rural areas [32]. Interventions particularly targeted to reach zero-dose children within these areas should also consider the social, economic, and cultural factors that influence health system deliveries in these areas. 

#### 3.2.2. Interventions for Community Engagement 

Similar to urban slum communities, key informants stressed how vaccination interventions should engage with diverse remote and rural communities. Community participation generates demand and accountability for vaccination, as well as other health services, and helps adapt the interventions to local contexts [32]. This concept was exemplified in two Indian states, where drum beating was used as a culturally appropriate, low-cost, and scalable method to alert communities about vaccination days for children in remote populations [33]. Engaging community leaders, particularly religious leaders as previously described, would also be key to reducing the number of zero-dose children in remote and rural settings. 

#### 3.2.3. Interventions for Health Systems Strengthening and Integration 

Integrating health services to enhance trust in vaccinations and health systems can especially help amplify the efforts to reach zero-dose children in remote or rural communities. My Village My Home (MVMH) is a community-level tool used in remote areas of India, Malawi, and Timor-Leste to record the births and vaccination dates of every infant in a community [34]. This poster-sized material helps community-level workers visually assess the vaccination status of all infants born within a year and to easily identify those who remain unvaccinated. In Ethiopia, “family folders” are paper based and, therefore, a more affordable system of tracking all of a family’s healthcare services and flagging those requiring follow-up, like immunization. Keeping these has helped healthcare workers obtain comprehensive data to monitor these childrens’ health needs and easily reach their patients and families to reduce losses to follow-up [35]. Within the routine health system monitoring, electronic immunization registries allow disaggregated data of registered children to understand population sizes and rack resources [32]. Reach Every District (RED) is another micro-planning tool to improve vaccine access for hard-to-reach populations by encouraging the building of micro plans to map barriers and identify solutions [36].

Some examples from non-health sectors in remote rural settings can be taken as inspirations to drive zero-dose vaccination uptake. These initiatives are largely based on integration with existing structures, addressing broader structural determinants. For example, in Chad, vaccination services are integrated with animal health services where combined human and animal health services achieved higher coverage among pastoralist communities. The program was found to be more cost-effective, feasible, and culturally acceptable than human services alone [37]. Similarly, health interventions integrated with agricultural services have been widely used. The OneAcre Fund delivers agricultural supplies in conjunction with hygiene products and COVID-19 vaccines in last-mile communities throughout sub-Saharan Africa [38]. Digital Green in India targets farmers and agricultural productivity by creating educational videos on safe motherhood and family planning with the community. These lessons can be expanded to include vaccine literacy to reach zero-dose children in remote and rural areas [39]. 

Finally, leveraging commercial partners in delivering vaccines could be a potential solution to reach otherwise isolated areas. For example, Coca-Cola’s supply chain has been successful in delivering lifesaving drugs in rural Tanzania and other African countries [40]. This “Last Mile” project came about after realizing that Coca-Cola reaches areas where the medical supply chain does not; areas that may also be left out of the reach out routine immunization programs. While it is currently used to deliver anti-malaria medications, the cold chain and supply infrastructure has the potential to be used for vaccine delivery as well. Tech Mahindra in India, which started as an agricultural equipment company, has committed to improving the global supply chain of vaccines, which could also be further leveraged to reach zero-dose children [41].

#### 3.2.4. Interventions for Technological Innovations in Vaccine Delivery

Contemporary technological solutions for real-time monitoring commonly include geospatial mapping that maps houses to target populations and augment mass vaccination campaigns. Reveal, a geospatial modeling platform with aerial satellite maps was used to identify built structures and locate zero-dose children in Zambia. Reveal uses smart maps and technology optimized for low-resource settings to monitor the coverage of interventions as they happen. This program supports decision makers and intervention managers by guiding and tracking the delivery of in-field activities and identifying responsible teams for follow-up [42]. 

Innovations in delivery mechanisms such as making vaccines safe, less wasteful, and minimally resource-intensive can potentially help in improving coverage and reaching zero-dose children. Remote temperature monitoring devices, such as ColdTrace5^®^, allow for the real-time monitoring of vaccine fridge temperatures and power availability, which have been installed in 120 vaccination sites in Tanzania and 36 sites in Kenya [43,44]. Microarray patches for thermostability and solar direct-drive refrigerators for enabling the conditions to store vaccines in electricity-deprived rural areas are examples of alternative modes of delivery [45,46]. The use of drones for vaccine delivery arose frequently in our review materials; drones have been used in countries, including Ghana, though our key informants were wary of the “jab and leave” mentality that such technology use may enable [47]. Our key informants reiterated that this process may leave people vaccinated but with no other health resources, without bolstering trust in and lasting connections with the overall health system. 

### 3.3. Conflict Settings

#### 3.3.1. Context for Barriers 

Barriers that children living in conflict regions face include those faced by other communities, but additionally include unique barriers such as the disruption of physical infrastructure and changing political trust. Conflict dramatically impedes vaccine coverage; for example, DTP3 coverage in Syria went from 80% in 2010 to 47% in 2018 after conflict erupted [48]. Diseases spread rapidly in conflict-affected regions because of a lack of infrastructure, close living spaces for displaced peoples, and climate factors. In conflict settings, unique barriers to vaccine delivery include supply chain disruptions, difficulty retaining health workers, insecurity, distrust between authorities and communities, as well as population displacement and migration. Many of the interventions discussed previously for urban and rural settings such as monitoring systems, cold chain enhancement, and drones can also be adapted to reach conflict settings. Below we focus on additional examples that specifically address the various interventions attempted to overcome the disruption of services specifically for conflict settings. 

#### 3.3.2. Interventions for Community Engagement

Part of what makes zero-dose children hard to reach in conflict settings is the added risk that working in conflict settings poses to healthcare workers. The loss of healthcare personnel in times of conflict can be drastic, though they are essential for successful vaccine delivery. An innovative approach to monitor the safety of healthcare workers is using WhatsApp messages or anonymous online reports to a hub to regularly share real-time damage to health facilities and threats against staff. This approach was implemented by ReliefWeb during the conflict in Syria to document violence against healthcare centers, specifying incident location, casualties, and facility affected [49].

In addition to planning for safety and negotiations, our key informants offered examples of incentivizing healthcare workers to provide compensation for the additional risk and fear that workers might encounter in conflict settings. This could be carried out with direct cash transfers, including via mobile money networks which may function even in fragile settings, as has been successfully implemented by the government of Liberia [50]. 

#### 3.3.3. Interventions for Health Systems Strengthening and Integration 

In conflict settings, some zero-dose children may not be registered with the health system and therefore would lack a vaccination card or even a birth record. The tracking and monitoring of zero-dose children have even more profound barriers for vaccine implementers in conflict settings. To address this issue, some tested interventions include Village Reach’s Electronic Immunization Registries (EIR) program and the International Red Cross’s mReach tracing data platform which enable health workers to register children and track their vaccination status [51,52]. mReach, coupled with Google Maps, was used in Somalia and the International Red Cross additionally used Digital Health ID along the Thai and Myanmar border. 

Disruptions in data sharing and misinformation are quick to take hold in conflict settings; hence, data and monitoring are critical not only for individuals but also for facilities. Therefore, strong monitoring systems help streamline supply chain demands and prevent the escalation of stockouts. The World Health Organization (WHO) has been using the Health Resources and Services Availability Monitoring System (HeRAMS) to monitor health systems and facilities functioning in various geographies which has been found to be especially helpful in conflict settings. In 2022, HeRAMS was used in 25 countries including many experiencing conflict, like Afghanistan, Iraq, the Central African Republic, and Ethiopia to name a few [53,54].

Leveraging other non-vaccine-specific humanitarian pathways can provide additional mechanisms to reach zero-dose children in conflict settings. For example, the World Food Programme is well equipped and experienced with a global logistics capacity in far-reaching geographies and even conflict regions [55]. Their supply chain has recently been used to deliver COVID-19 vaccines to Timor-Leste, for example, and has therefore demonstrated the feasibility of utilizing this humanitarian transportation network for vaccines in the future [56]. Aid workers that have access to areas with children in high need of care should be equipped to also provide lifesaving vaccines if they are not already. Because children experiencing malnutrition, often heightened during conflict, are at higher risk of dying from vaccine-preventable illnesses, they are a key population to target in vaccination efforts. Thus, leveraging existing humanitarian pathways may help reach zero-dose children. 

#### 3.3.4. Interventions for Technological Innovations in Vaccine Delivery

An innovative technology being used to improve monitoring and data tracking in conflict settings is Biometrics [57]. Biometrics uses fingerprints or iris scanning as a key for aid distribution and tracking migration, and its iRespond system has previously been used in Myanmar, Senegal, and Sierra Leone [58]. Biometrics have also been used for food distribution by the World Food Program in refugee camps in Kenya and Jordan, and can be utilized to track the vaccination status of zero-dose children [58,59]. Biometrics technology may be useful in tracking displaced zero-dose children.

## 4. Discussion

Through this narrative review, we have identified possible interventions within the health and non-health sectors that can be employed to facilitate the immunization of zero-dose children, particularly in LMICs. These interventions apply to three priority scenarios, which include urban slums, remote or rural communities, and conflict zones. These interventions fell into three thematic areas for each of the three settings: (1) community engagement, (2) health systems’ strengthening and integration, and (3) technological innovations. 

We made a significant effort to identify interventions that have already been used to reach zero-dose children and novel solutions and that may be used, including some that do not yet have evidence for their implementation. Nonetheless, it is possible to anticipate the outcomes and challenges that may arise with the implementation of such interventions. For example, community engagement interventions would be anticipated to have an impact on creating vaccine demand and decreasing hesitancy as they are aimed at increasing awareness through messaging, building trust in vaccination through community leaders and religious figures, and increasing knowledge of vaccine availability [18,24]. Conversely, community engagement interventions would do little in terms of supply chain optimization, and these activities may face backlash from those who are opposed to vaccination. Interventions under health systems’ strengthening can be expected to improve the supply and distribution of vaccines to underserved areas, however, there is a greater cost associated with these interventions and coordination that may require existing health infrastructure and staffing [32,36]. Technological interventions are gaining popularity and can have positive outcomes in aspects of distribution, monitoring, and evaluation, though barriers to these interventions would include cost, digital literacy, and connectivity [57,58]. As particularly highlighted by our key informants, these interventions may have significant potential for impact, but if implemented alone, they can significantly undermine the fundamental gaps that community engagement and health systems’ strengthening interventions aim to address.

Overall, the evidence and key informants suggested that due to various socio-cultural, economic, and political factors that hamper efforts to vaccinate zero-dose children in different settings, diverse interventions are required to overcome those barriers. Interventions should be nuanced in an effort to overcome context-specific barriers. Some interventions cater better to those who refuse to get vaccinated due to distrust of the health system, religious beliefs, or other reasons [16,17,18,19]. Others cater to those who have very little access to primary care services, as zero-dose children face structural barriers to most health services. In many ways, the point that interventions should be identified and adapted based on individual contexts seems obvious, yet we were struck by the emphasis on this both from the review materials and the key informants. This emphasis likely signals an overreliance on “one size fits all” interventions supported globally to reach zero-dose children [19,60].

Furthermore, while innovative technological interventions may ensure that vaccines can reach zero-dose children, they do not guarantee an increase in vaccine uptake in and of themselves. For these interventions to be effective and sustainable, they must be supplemented with initiatives that aim to improve vaccine acceptance in communities and integrate these interventions into the existing primary health care structures. Given the general critique of technological interventions, such as drones, causing harm and skepticism in addition to doing good [61], we suggest ensuring that such interventions are coupled with community engagement efforts and attempts to strengthen existing health systems rather than bypassing them. Additionally, working with existing community leaders and health workers helps mobilize community members and identify other issues that are a priority to the population, which, in turn, could improve vaccine uptake [62,63]. Monitoring and evaluating these interventions and programs can play a pivotal role in ensuring that zero-dose children are enumerated and vaccinated. 

As the global focus on vaccinating zero-dose children increases, our narrative review, the first of its kind to the best of our knowledge, is immensely timely and helpful. However, our review faces several limitations. First, there is a small evidence base for the effectiveness of most of the interventions that our review identified, despite enthusiasm for them, and, hence, our heavy reliance on gray literature. There is a need to evaluate if interventions to increase vaccine uptake among under-vaccinated children have been successful, and if these gains can be observed for reaching zero-dose children specifically. Second, there is limited evidence on the cost-effectiveness of scaling these interventions in specific contexts to support the implementation of these programs. Third, many of these interventions reach not only zero-dose but also under-vaccinated children and, therefore, the data evaluating how many zero-dose children have been reached is often unclear. Fourth, our approach has some methodological limitations: (1) a narrative review, by definition, is not easily reproducible and the reproducibility of our current work is further complicated by our inclusion of gray literature; (2) iterations or variations of the search string, e.g., the inclusion of “equity” alongside “inequity” may produce different search results, and our work did not attempt to be exhaustive in its approach. Notwithstanding these limitations, our current work lays out an overview of potential interventions from both health and non-health sectors to consider for reaching zero-dose children, especially in the three priority settings in LMICs.

## 5. Conclusions

Our narrative review identified three priority settings (urban slums, remote or rural communities, and conflict settings) and three thematic areas (community engagement, health systems’ strengthening and integration, and technological innovations) to focus interventions to reach and vaccinate zero-dose children. The barriers to reaching zero-dose children are multifaceted and nuanced to each setting, therefore, no one intervention is enough to extend the reach of vaccination for all the various settings. Technological interventions especially must be coupled with community engagement efforts and attempts to strengthen existing health systems rather than bypassing them. Future evaluations, including cost-effectiveness studies, of the suggested interventions are urgently needed to guide scale-up, as the evidence-base around these interventions is relatively small.

## Figures and Tables

**Table 1 viruses-15-02092-t001:** Examples of interventions to reach zero-dose children based on thematic areas in each setting.

Setting	Community Engagement	Health Systems Strengthening and Integration	Technological Innovations
Urban Slums	- Art for public health messaging (ex: GOAL Zimbabwe and M-pesa) - Community-based outreach - Utilizing religious leaders, i.e., Mobile Mullahs - Women support groups/Mother Meetings	- Incentives for CHWs/ASHAs - Slum health committees - Referral systems (ex: Roadmap for Achieving Universal Immunization Coverage) - Community mapping for timing and location (ex: Humara Bachpan) - Distribution of Vaccination Centers	- Monitoring and Evaluation (India’s urban immunization dashboard), Uganda’s National Adverse Event Following Immunization (AEFI) committee - GIS for community mapping
Remote/Rural	- Culturally specific messaging (ex: drum beating)	- Electronic Immunization Registers (ex: family folders) - My Village My Home - Reach Every District (RED) - Integration with agricultural, animal health, and commercial sector services (ex: Project Last Mile, OneAcre Fund, Digital Green)	- Remote temperature monitoring devices (ex: ColdTrace5, Microarray patches, and solar direct-drive refrigerators) - Drone Delivery - Geospatial monitoring (ex: Reveal)
Conflict Zones	- Increase healthcare workers’ communication and access to information through WhatsApp messaging or anonymous online hubs - Incentivize healthcare workers to acknowledge the risk of working in a conflict zone	- Electronic Immunization Registers - IRC’s mReach tracing data platform - Digital Health IDs - Monitoring facilities (WHO’s Health Resources and Services Availability Monitoring System) - Integration with other humanitarian response services (ex: World Food Program)	- Biometrics (ex: iRespond)

## Data Availability

No new data were generated for the review itself. The key informant data presented in this study are available on request from the corresponding author. These data are not publicly available due to potential risk of loss of confidentiality.

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
