# Peer review of "Interventions to Vaccinate Zero-Dose Children: A Narrative Review and Synthesis"

_viruses, 2023, doi:10.3390/v15102092_

Round 1

Reviewer 1 Report

The article focuses on and reviews a very important topic of children vaccination.

However, the method of work and methodology used are not clear. The authors claim that they are doing 'narrative review'. However, the article does not work with narratives and does not do narrative review/analysis. 

The authors work with two types of data and tools : they combine literature analysis of the sources written on the topic with 6 in depth interviews with academics and key informants working in the field of vaccination. 

There is a place to explain the methodological approach chosen, to explain the rationale of combining  the two types of data - literature review and in- depth interviews. There is a place to explain the sampling rationale: number  and choice of in-depth interview participants. It should be also explained how text analysis (interviews texts ) was done. Please clarify why the interviews were not recorded. 

In addition. it is advised to check the logics of presentation of the main themes. 

Please also see the pdf file with short comments and questions.

Reviewer 2 Report

I am grateful for the kind invitation to undertake a critical review of the manuscript titled "Interventions to Vaccinate Zero-Dose Children: A Narrative Review and Synthesis." This manuscript addresses a topic that holds utmost significance within the realm of public health, particularly in those regions where the accessibility to vaccines is hampered by specific contextual factors.

In my thorough examination of the manuscript, I have encountered only one minor point that I wish to draw the authors' attention towards. My recommendation centers on the need to expand upon the discourse pertaining to the proposed interventions. While the manuscript does commendably mention various interventions and underscores the significance of adopting a comprehensive and individualized approach, it falls short in providing an in-depth exploration of each intervention. Consequently, I propose a brief but more comprehensive elaboration of the discussion, placing particular emphasis on elucidating the anticipated outcomes and the potential challenges that may arise during their implementation.

Given my professional background in the field of public health, I found the manuscript to be both insightful and engaging. I would like to take this opportunity to extend my heartfelt congratulations to the authors for their commendable work. Your dedication to addressing this critical issue is evident, and I commend your efforts in advancing the discourse on vaccination strategies for zero-dose children.

Reviewer 3 Report

The search terms required a lot of components and likely excluded a number of papers. What about countries that refer to penta1 rather than dtp1? Did not include infants in first set of mesh terms. “Challenges, barriers, obstacles, disparity, inequity, and coverage” would not capture equity or identifying zero-dose children. In running a similar search over the 2010-2021 time period, came up with 10,000 articles.

If the target was only in LMICs why was this not part of the search terms? Is there a reason you limited to pubmed and didn’t consider other search databases?

Where is the list of articles included and how they were classified? It would be good to include this as a supplemental table, so readers can find the information behind the findings you describe.

There no flow diagram included to clarify what papers were excluded at which phase and for what reasons. This needs to be added to the manuscript.

It is unclear from reading the manuscript how the information from the 6 key informant interviews was used. Were these interviews then coded in some way? Did you use a specific framework? How was the number 6 decided? What types of questions were asked in the semi-structured interviews and how did this inform the results?

Please add to limitations a note about the inherent study design of a narrative review has a potential for bias because it is not comprehensive, only one database was used, and only 6 authors were interviewed. Other interventions may exist that are not captured here.

How does reference 40. Tanzania delivering medicines via Coca-Cola fit into the review? There is no indication from that reference that this was used to deliver vaccines, and does not specify reaching zero-dose or unvaccinated children.

This paper seems to mix two different concepts: evidence of interventions that have been used to reach zero-dose children and potential transference of interventions that have been used for other initiatives that could be leveraged to reach zero-dose. It presents as though it’s evidence-based, but many of these examples are opinion-based. For example, references to using the infant formula pathway (55 & 56) is a suggestion noting what happens with infant formula provision but does not include an example of how this has been used for vaccine delivery or if it’s even feasible. Vaccines require cold chain, and infant formula does not. This is more speculative and would not have come through in the pubmed search. Presenting all these interventions together gives the reader the notion that these interventions are all of the same caliber and feasibility. The ones that the authors are calling for use to reach zero-dose children should be clearly separated from those that have been used.
